# Precise Phenotyping for Improved Crop Quality and Management in Protected Cropping: A Review

Chelsea R. Maier [1,*], Zhong-Hua Chen [2], Christopher I. Cazzonelli [1], David T. Tissue [1,3] and Oula Ghannoum [1]

1   Hawkesbury Institute for the Environment, Western Sydney University, Richmond, NSW 2753, Australia
2   School of Science, Western Sydney University, Richmond, NSW 2753, Australia
3   Global Centre for Land Based Innovation, Western Sydney University, Hawkesbury Campus, Richmond, NSW 2753, Australia
*   Correspondence: chelseamaier@gmail.com

**Abstract:** Protected cropping produces more food per land area than field-grown crops. Protected cropping includes low-tech polytunnels utilizing protective coverings, medium-tech facilities with some environmental control, and high-tech facilities such as fully automated glasshouses and indoor vertical farms. High crop productivity and quality are maintained by using environmental control systems and advanced precision phenotyping sensor technologies that were first developed for broadacre agricultural and can now be utilized for protected-cropping applications. This paper reviews the state of the global protected-cropping industry and current precision phenotyping methodology and technology that is used or can be used to advance crop productivity and quality in a protected growth environment. This review assesses various sensor technologies that can monitor and maintain microclimate parameters, as well as be used to assess plant productivity and produce quality. The adoption of precision phenotyping technologies is required for sustaining future food security and enhancing nutritional quality.

**Keywords:** precision phenotyping; protected cropping; sensor technology; indoor agriculture; food security

## 1. Introduction

By 2050, the United Nations (UN) projects that that the world's human population will reach 9.7 billion, and an estimated two-thirds of the population will live in urban environments [1]. An increased population with the increased frequency of extreme weather events has made food security a critical topic of global concern [2]. The UN Food and Agriculture Organization (FAO) has identified three key areas to increase agricultural food security. Two of these key areas are to increase the intensity of farming and to increase innovative technology to make farming more efficient [3].

Protected cropping is a cultivation method that greatly improves yields by more intensively producing crops on less land area and is defined as a horticultural growing process that utilizes a structure to protect crops from abiotic and biotic stressors. Protected cropping is also known by other names such as protected cultivation and protected agriculture [4]. Protected cropping usually involves various degrees of environmental control (e.g., temperature, humidity, $CO_2$, and nutrition) to reach the highest yields possible and includes a range of facilities from low-tech shade houses to completely climate-controlled high-tech glasshouses [5,6] Compared to broadacre agriculture, protected cropping improves crop yields and quality, and resource efficiency; increases resiliency to extreme weather events; and, thus, increases national food security [7]. Protected cropping also has the ability to provide food supply year-round independent of the seasonal changes in environmental parameters that drive crop productivity.

Technology across the production chain in the protected-cropping industry is identified as a main influential factor in improving product quality and outcome and is regarded as

essential for future food security [7,8]. These include phenotyping technologies that monitor genotypic expression of plant traits for a suite of goals, such as to inform breeding programs and plant-health assessments. Phenotyping technology has been developed primarily for field and broadacre agriculture through precision phenotyping [9,10]. However, these methods can be adapted and adopted for use in protected-cropping facilities. The uptake of these technologies is not yet widespread due to capital investment, added maintenance costs, education about using new technologies, and technology transfer from traditional cropping to the protected-cropping industry [7,11]. The uptake of climate monitoring and control technology will increase productivity within protected-cropping structures already in existence, and this is part of the key objectives to expand the protected cropping industry [7]. The application of precision phenotype monitoring in protected cropping will further increase yields and crop quality across the full range of protected-cropping-facility types.

This review will serve to inform about the current state of the protected-cropping industry, precision phenotyping technologies, and associated climate-monitoring/control technologies required to phenotype plant growth and productivity and integrate disease management. The review also emphasizes crop quality improvement and the need to bolster food security in an energy-efficient manner.

## 2. Overview of Protected-Cropping Advantages and Areas of Expansion

Protected cropping is a rapidly expanding sector of horticulture throughout the globe, with China currently having the largest surface area of greenhouses in the world [12]. In 2014, it was estimated that 5 million hectares are under protected-cropping facilities [12]. Ninety percent of protected-cropping facilities are located in Japan, China, and Korea, and the other ten percent are distributed across Europe, Africa, the Middle East, North America, Central and South America, and Oceania [12]. Protected-cropping greenhouses produce 60% of the fresh vegetables globally; however, the total area they comprise is only 400 km $\times$ 100 km. While individual protected-cropping facility locations present specific climate considerations, overall protected cropping is distributed across three major climate regions, which are Mediterranean, Temperate, and Subtropical/Tropical, with the majority being in a Temperate region (90%) [12].

The FAO and Organization for Economic Co-operation and Development (OECD) have identified that critical drivers to increase agricultural productivity, food security and sustainability are investment in technology, infrastructure, and agricultural training [3]. With the continued transition to more intensive production methods, 87% of the projected growth of global crop production over the next 10 years will come from yield improvements through more intensive agriculture and advancements in technology-assisted farming, while only 6% is projected to come from the expansion of cropland [3]. Protected cropping is of particular interest in regions of the world that are arid and semi-arid, such as Australia and Gulf Cooperative Council (GCC) countries, where water is limited, as protected-cropping methodology can reduce water usage and increase productions up to five-fold (Figure 1) [12–15].

While being more productive per unit of land area and using less fertilizer and water than field-grown crops, plants grown in protected-cropping facilities require more energy per kilogram of produce and higher skilled labor than do field-based production systems [16,17]. The protected-cropping industry is researching ways in which to reduce energy consumption and labor costs. Solutions will partly come from increased automation (e.g., harvesting) and real-time phenotyping linked to crop-management decision-making [18,19]. Non-destructive phenotyping is used to monitor crop health and fruit quality to inform about appropriate environmental control, management strategies, integrated disease and pest control, and plant and fruit nutrient status for optimal-quality horticultural production. Whilst some technologies are already available for producers, much work remains to be performed to adapt and fine-tune sensing and imaging tech-

nologies to protected-cropping industry and conditions and to develop cost-effective and easy-to-use alternatives.

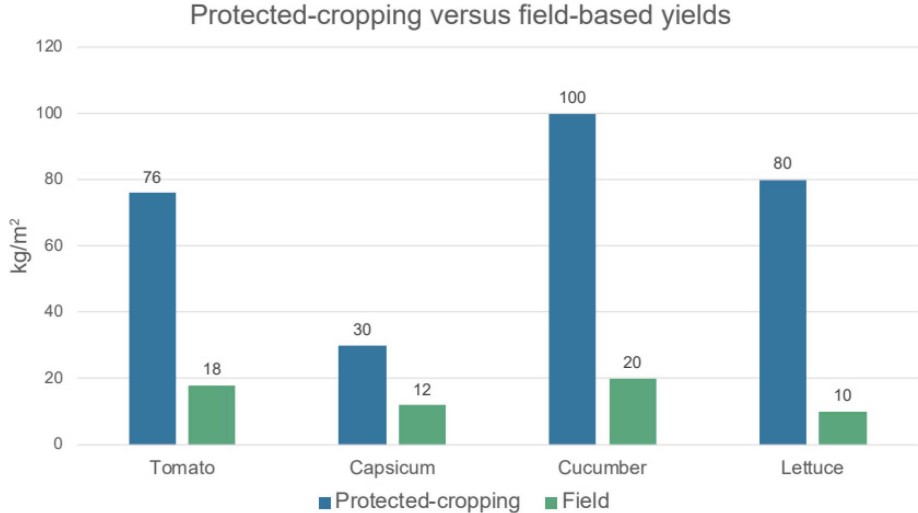

**Figure 1.** Example data of Australian crops grown by using protected-cropping methods versus field-based crops. Data from Smith (2011) [13].

## 3. Using Phenotyping to Advance the Protected-Cropping Industry

A crop phenotype is the observable characteristics that are expressed as a result of a genotype's interaction with the environment. Phenotyping is the science of the characterization of the crop's traits such as growth, development, architecture, physiology, health, nutrition, and yield. While many plant phenotyping techniques have been developed for the purpose of breeding, disease detection, and improving stress tolerance for field crops [20–22], phenotyping is also used to monitor plant health, development, disease manifestation, and fruit quality so that growers can take early action to change abiotic conditions and maximize the yield of crops. Crop phenotyping can optimize breeding programs for new varieties that will produce food of high nutritional and aesthetic quality [23]. With appropriate phenotype monitoring, resource-use efficiency can be maximized [24,25].

Crop phenotyping is an essential process in vegetable and fruit production and is particularly important to mitigate the impacts of climate change, as new varieties of heat- and drought-tolerant fruit and vegetables will have to be developed in the future to maintain crop yield and quality. Currently, assessment of phenotype characteristics relies largely on visual scoring methods by experts, which is time-consuming and can introduce bias via human error [23]. Thus, plant phenotyping has become a field for innovation to hasten breeding programs and quality assessment, pre- and post-harvest. High-throughput phenotyping (HTP) platforms have been developed and are being used to collect data for quantitative studies of complex traits related to the growth, yield, and adaptation to biotic (e.g., disease and insects) and abiotic (e.g., drought and salinity) stress. There are a number of techniques available or in development to assess these traits; however, to properly assess phenotypic expression, it is also important to measure and control the environmental parameters to which a crop is exposed, as phenotypic expression is highly dependent on environmental conditions. While much of the tools used for precision phenotyping were developed for broadacre crops, these technologies are being transitioned to protected-cropping facilities in various ways [10]. Once systems are developed to streamline precision phenotyping in protected cropping, these data-collection systems can be further integrated into automated control systems that will be able to autonomously make crop-management decisions based on the observed plant expression [4,26].

### 4. Environmental Monitoring Is a Prerequisite for Plant Phenotyping

*4.1. Phenotypic Plasticity in Response to Environmental Parameters*

In addition to crop-related traits, effective phenotyping requires continuous assessment of environmental parameters. The environmental conditions in which a crop is grown must be known and, ideally, controlled for consistency across a crop cycle. The primary environmental conditions that impact crop growth are root-zone temperature, moisture and electrical conductivity (EC), light quality and quantity, air temperature, and relative humidity (RH). While some impacts of these environmental parameters on external and internal phenotypic expression are known [5], the field is being explored further as better methods of environmental and phenotypic monitoring become accessible.

*4.2. Root-Zone Temperature, Moisture Content, and Electrical Conductivity*

Root-zone temperature, moisture and EC play key roles in crop performance. While controlling root-zone temperature is a by-product of maintaining optimal air temperature, substrate moisture and EC are controlled by irrigation and fertilization regimes. Irrigation timing impacts root development in soils and soilless substrates. A moisture-saturated root zone results in poor root development, and if this development is hindered early in the growth of the plant, that plant's ability to produce and bear fruit is reduced because the root structure is not there to maintain it. In the case of hydroponic cultivation, EC is an indirect measure of nutrient availability and is crucial for nutrient uptake, and optimal EC is highly specific to the crop. EC that is higher than necessary results in stagnation of nutrient ions. High EC has been related to blossom end rot in tomatoes, as, with increasing EC, mass flow is reduced due to less water entering the plant roots [27]. Very low EC, in the case of hydroponic crops, can lead to root cell rupture due to the large imbalance of osmotic pressure across the root membrane. An example of sensor technology that measures root-zone EC, moisture, and temperature can be found in Table 1. Some root-zone temperature, moisture, and EC sensors can be integrated into sophisticated greenhouse and glasshouse control systems to dictate irrigation cycles to optimize plant access to water and nutrients throughout the crop's growth cycle [28].

*4.3. Light Quality and Quantity*

Light quality and quantity impact plant development, as light is the main driver in crop production. It is estimated that light quantity is reduced by 30% in glasshouses due to structural shading; this paired with low light intensity of the winter months has been linked to low yields and the production of small fruit [29]. The protected-cropping sector has developed cladding material specifically created to not block solar radiation, as the whole spectrum of solar radiation plays an important role in plant development. Too little natural light leads to elongation of stems and low fruit yield. In cucumbers, fruit grown in low light conditions tends to be lighter in color and yellow more quickly once harvested [30]. Misshapen, swollen, and hollow tomato formations have also been documented as a result of low light conditions [29]. In contrast, a higher irradiance from the sun results in leaf dehydration and premature shutdown, leading to a reduced photosynthetic capacity and lower crop yield. Excess light can lead to sunscald in a wide range of crops, including tomatoes and capsicum [29]. In lettuce, high light environments can lead to the increased presence of tip burn [31]. Light quality also influences internal and external phenotypic expression. Plants primarily use photosynthetically active radiation (PAR), which consists of wavelengths from 300 nm to 700 nm. Red light exposure has been shown to reduce the bitter flavor in lettuce leaves [32], while UV-B exposure causes the accumulation of secondary metabolites, which influence a variety of plant physiological processes and internal quality of some vegetables [29]. In protected cropping, PAR sensors or global radiation sensors are used both outside and inside the growth compartments to monitor light levels (Table 1). These measurements can be and often are integrated into shade curtain control systems to ensure that crops are receiving appropriate light conditions throughout the day [33].

Light quality also plays a key role in taste and aroma profiles for different fruiting or vegetative plants. Light quality, intensity, and duration can be manipulated to elicit desired attributes of leafy greens, including leaf shape, leaf color, nutrition, flavor, texture, and aroma [34].

### 4.4. Temperature and Relative Humidity (RH)

Temperature and RH are important environmental variables that must be maintained for proper crop development and to achieve maximum yield. With an increase in temperature, RH decreases, as RH is a measure of the percentage of moisture the air can hold as vapor at that particular temperature. As temperature increases, the air's ability to hold more moisture increases. Crops also have different microclimate needs at different stages of development; thus, continuous measurement is necessary to maintain these parameters, which can be performed with a variety of sensors, an example of which can be found in Table 1. While these temperature and RH sensors are used to continuously monitor the microclimate, they are also the main trigger for misting/dehumidifying and heating/cooling systems to maintain optimal RH and temperatures, and they are essential tools for maintaining high crop yield and quality [35].

While optimal temperatures for day and night will vary with crop type and variety, the majority of horticultural crops are warm-weather plant species, with their main growing seasons occurring over the summer months. Typically, temperatures between 20 and 30 °C and with an RH of 50 to 80% are desirable; however, each crop variety has specific requirements. Cucumbers, for instance, have been shown to taste sweeter when grown in moderate RH as opposed to high RH and, in general, produce better fruit with respect to quality attributes and external appeal [36]. While air temperature is an important parameter, studies have shown that canopy (leaf) temperature may be far more important. Leaf temperature is directly related to transpiration rates and measurement of this parameter provides information about leaf temperature, as well as plant water status [37].

### 4.5. Sensor Technology to Monitor Environmental Parameters

Consistent environmental conditions are essential for reproducible phenotypic responses from different genotypes. Therefore, if environmental conditions are tightly controlled, ideal plant growth and development can be achieved; thus, developmental stages can be identified and targeted to identify specific traits. Moreover, in tightly controlled conditions, deviations from typical phenotypic expression become more obvious and make assessing disease, plant health, and fruit quality more straightforward. There are various sensors available to monitor air temperature, RH, root-zone temperature, moisture, and EC, as well as light quantity and quality (Table 1). While all of these parameters, such as shade curtains or irrigation, can be monitored individually via computer or smartphone and the control mechanism changed manually, there are integrated hardware and software systems available that can automatically manage these environmental controls. Due to the variability of microclimates within protected-cropping facilities, high-resolution measurements are useful for correctly informing the grower on the overall environment that the crop is experiencing; thus, they can make more appropriate management decisions [38,39].

**Table 1.** Environmental parameters and crop impact, listed with available sensor technology and control mechanisms to maintain optimal conditions. Photo credits: (1) www.grodan.com (accessed on 14 September 2022), (2) www.vaisala.com (accessed on 14 September 2022), (3) www.apogeeinstruments.com (accessed on 14 September 2022), and (4) www.directindustry.com (accessed on 14 September 2022).

| Environmental Parameter | Impact on Crop | Sensor | Control Mechanism | Example |
|---|---|---|---|---|
| Electrical conductivity (EC) | High: Blossom-end rot, nutrient deficiency, and reduced yield. Low: Cell rupture. | Slab or soil EC sensors (usually include temperature and moisture measurements) | Irrigation regimes, pH modification, and EC modification of stock solution | |
| Root-zone moisture | High: Roots do not develop enough to support a full-grown producing plant. Low: Root die-off and plant dehydration. | Soil-moisture probes or slab or soil EC sensors (usually include temperature and moisture measurements) | Properly timed irrigation and proper landscaping to prevent pooling (slope) |  |
| Root-zone temperature | High: >25 °C, $NH_4$ toxification, leading to cell death. Low: 3–11 °C, $NH_4$ uptake stimulates plant growth. | Soil-temperature and moisture probes that include EC measurements | Shade cloth, irrigation solution temperature, heating pad, and heating cables | |
| Air temperature | High: Leaf dehydration and earlier stomatal shutdown. Metabolic shutdown due to inability to dissipate heat. Low: Delayed blooming and stunted or slow growth. Large day–night temperature differentials impact fruit set. | Dual air-temperature and relative-humidity probes | Pad and fan cooling, cold-coil fan cooling, shade cloth to reduce radiant heat, hot-water pipes, and hot air via external heat source |  |
| Relative humidity | High: Low stomatal conductance, reducing nutrient distribution to plant and fruit. Low: Early stomatal shutdown, resulting in reduced photosynthesis. | Dual air-temperature and relative-humidity probes | Misting system, condensing system, and dehumidification | |
| Light quality | 280 nm: Reduces quantum yield and rate of photosynthesis. 315–400 nm: Promotes pigmentation and thickens plant leaves. 400–440 nm: Promotes vegetative growth. 640–660 nm: Vital for flowering. 740 nm: Increases photosynthesis [40]. | Spectroradiometer or a combination of PAR and net radiometer | Colored shade cloth, fluorescent films, and light supplementation |  |
| Light quantity | High: Leaf dehydration, sunscald, photodamage, and lowered photosynthetic rates. Low: Stem elongation, lower photosynthetic rate, reduced yield, misshapen fruit, and reduced shelf life. | PAR sensors | Shade cloth and light supplementation with light-emitting diodes |  |

*4.6. Environmental Monitoring Is a Prerequisite for Plant Phenotyping*

In summary, plants have plasticity to alter their phenotype, exhibiting different external and internal expression depending on the environment. Environmental control and data capture are important throughout the phenotyping process, as antecedent events can be inherited by the progeny, which is known as the 'memory effect' [41]. Frequent high-throughput phenotyping will promote faster phenotypic data acquisition for correlation with genomic information [42]. In protected cropping, phenotypic expression dictates fruit aesthetic and nutritional quality pre- and post-harvest, suggesting that two important subjects need to be considered:

- Precise control over crop microclimate to maintain desired phenotypic expression across crop cycles;
- Frequent phenotypic surveys of plants and fruit, throughout the cropping cycle and during post-harvest sorting, storage, and distribution.

## 5. Non-Destructive Phenotyping in Protected Cropping

*5.1. Overview*

In the past, plant phenotyping techniques have required destructive measurements in order to assess plant health, fruit quality, and the presence of pests or disease. However, with the advancement of optical sensors, gas chromatography, and other optical analytical methods, plant phenotyping can be performed in real time, non-destructively. Imaging techniques with computer analysis provide fast and non-destructive methods by which to evaluate fruit during their development, harvest, and post-harvest periods. The use of these technologies started in the 1990s after the development of charge-coupled-device (CCD) and complementary-metal–oxide–semiconductor (CMOS) technologies [41]. CCD and CMOS sensors are used to measure color in different food products, from seed to fruit quality [43]. These applications are increasingly being used for fruit quality control.

Phenotypic imaging techniques span the electromagnetic spectrum and include machine-vision visible imaging, imaging spectroscopy (multispectral and hyperspectral remote sensing), thermal infra-red imaging, fluorescence imaging, 3D imaging, and tomographic imaging (magnetic resonance, positron emission, and computer tomography) (Table 2). Today, imaging plants is more than taking photographs with Red, Green and Blue (RGB) cameras; it also includes the precise measuring of the wavelengths of photons reflected, absorbed, or transmitted by the plant tissue. Each component of plant cells has wavelength-specific transmittance, absorbance, and reflectance properties [23]. Primarily, visible imaging techniques are used to measure plant architecture, such as biomass, leaf area, color, growth dynamics, seed vigor and morphology, and root architecture, as well as leaf disease, yield, and fruit number and distribution. Disease can be detected by the use of fluorescence imaging. Plants' temperature and stomatal conductance can be measured by thermal infra-red imaging and are related to plants' water status and transpiration rate [23].

Other techniques include a microwave resonator, which can be used to non-invasively determine water content and then interpolate the total plant biomass. The dielectric properties of a microwave resonator change when plant material is inserted into the cavity, and this change is proportional to plant water content. By separating the root and media from the plant with a copper plate, it was possible to monitor intact plants and assess diel growth patterns, allowing for the fast integrative assessment of plant growth, water status, and physical attributes. This is an important metric to be able to assess because a plant's ability to produce biomass determines plant vigor and eventual crop yield [44].

Gas chromatography with mass spectrometry and proton transfer reaction–mass spectrometry are used to identify and quantify volatile organic compounds (VOCs) emitted by a plant (Table 2). The VOC profile from a plant is dependent on its stage of growth; this profile also changes if the plant is experiencing stress from biotic and/or abiotic sources. Spore detection methods are also being used as an early detection system for disease. While there are techniques for microbial identification that require culturing, methods are being developed that allow for the real-time identification of fungal spores through the use of

optical analysis [45]. The external evaluation of the fruit and edible portions of commercial crops is essential for marketability, further trait selection, and the development of proper crop-management practices. As all data must be comparable [41], standardization should be a key aspect of sensor development across the industry as these techniques progress.

**Table 2.** Summary of phenotyping techniques and their applications. Adapted from Li et al. (2014) [23]. Photo credits: (5) https://fluorcams.psi.cz/ (accessed on 14 September 2022), (6) www.mouser.com (accessed on 14 September 2022), (7) www.canr.msu.edu/ (accessed on 14 September 2022), (8) https://www.middletonspectral.com/ (accessed on 14 September 2022), (9) https://voltrium.wordpress.com/ (accessed on 14 September 2022), (10) www.faro.com (accessed on 14 September 2022), and (11) www.pmeasuring.com/ (accessed on 14 September 2022).

| Phenotyping Technique | Sensor | Resolution | Phenotype Parameters | Examples |
|---|---|---|---|---|
| **Imaging Techniques** | | | | |
| Visible-light imaging | Cameras sensitive in the visible spectral range | Time series of whole organ or organ parts | Shoot biomass, yield, root architecture, germination rate, morphology, height, size, and flowering time | |
| Fluorescence imaging | Fluorescence cameras and setups | Whole shoot or leaf tissue; time series | Photosynthetic status (variable fluorescence), quantum yield, leaf health status, and shoot architecture |  |
| Thermal imaging | Near-infra-red cameras | Pixel-based map of surface temperature in the infra-red region | Canopy or leaf temperature; insect infestation of grain |  |
| Near infra-red imaging | Near-infra-red cameras; multispectral line scanning cameras; active thermography | Continuous or discrete spectra for each pixel in the near-infra-red region | Water-content-composition parameters for seeds; leaf area index |  |
| Hyperspectral imaging | Near-infra-red instruments and spectrometers, hyperspectral cameras, and thermal cameras | Crop vegetation cycles and indoor time-series experiments | Leaf and canopy water status, leaf and canopy health status, panicle health status, leaf growth, and coverage density |   |
| 3D imaging | Stereo camera systems; time-of-flight cameras | Whole-shoot time series at various resolutions | Shoot structure, leaf-angle distributions, canopy structure, root architecture, and height |  |
| Laser imaging | Laser-scanning instruments with widely different ranges | Whole-shoot time series at various resolutions | Shoot biomass and structure, leaf-angle distributions, canopy structure, root architecture, height, and stem length | |
| **Gas and VOC analysis** | | | | |
| Proton transfer reaction–mass spectrometry | Mass spectrometer | Whole plant or single leaf | Pest presence, abiotic stress indicator | |
| Gas chromatography with mass spectrometry | Mass spectrometer | Whole plant or single leaf | Pest presence, abiotic stress indicator | |

**Table 2.** *Cont.*

| Phenotyping Technique | Sensor | Resolution | Phenotype Parameters | Examples |
|---|---|---|---|---|
| **Fungal detection techniques** | | | | |
| Impinger or wet-cyclone | Liquid entrainment for optical analysis | Depends on entrainment method | Size, scatter, and pigmentation | |
| Wide issue bioaerosol spectrometer (WIBS) | Optical sensors | 0.8–20 µm | Particle size, symmetry, scatter, fluorescence, and absorbance | |
| Particle fluorescence | Optical sensors | 0.5–50 µm | Particle fluorescence | |

### 5.2. Crop Growth and Yield

Thermal infra-red imaging can be used to measure crop canopy temperatures and assess plants' water status (Figure 2). Due to evaporative cooling, this is the main influence of leaf temperature, and there is a direct relationship between stomatal conductance, transpiration rate, and leaf temperature. However, there are technical challenges in adopting this method, as environmental temperature and air movement can impact the measurements [46]. RGB/visible measurements are used to assess the growth rate or biomass accumulation. Near-infra-red measurements can be used for decreasing leaf water content [47].

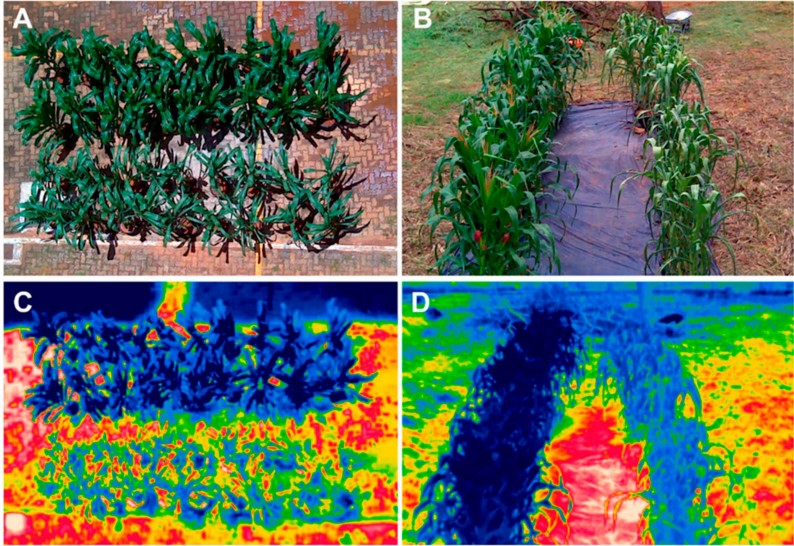

**Figure 2.** RGB (**A**,**B**) and thermal images (**C**,**D**) of maize plants taken on the 4th (**A**,**C**) and 12th (**B**,**D**) day of drought stress. The upper (**A**,**C**) and left-hand-side (**B**,**D**) rows of dark green (RGB) and dark blue (thermal) are control plants, while the lower and right-hand-side rows of pale green and light blue are drought-stressed plants [48]. Reproduced with permission from Carlos A. F. Sousa, International Journal of Molecular Sciences; published by MDPI, 2019.

### 5.3. Fruit and Leaf Quality

Hyperspectral imaging for crop-quality assessment was recently developed. Fruits contain different concentrations of nutrients depending on the environmental conditions. These different internal chemical compositions scatter, reflect, absorb, and/or emit different wavelengths of electromagnetic energy in specific ways, and, thus, light can be used to non-destructively characterize the fruit and other organic components of a plant. Foliar developmental characterization, fruit-stage determination, and stress or disease presence can be assessed once spectral characteristics for these attributes have been calibrated. An example of how these spectral responses are shown via false color imaging is given in

Figure 3. Fruit and leaf stress can be identified by numerous imaging techniques; however, there are multiple causes for stress. They could be abiotic or biotic in nature, and, therefore, modeling for particular stresses needs to occur. Applying spatial patterning analyses to multispectral and hyperspectral imaging will greatly improve the determination of the cause of stress and thus aid in the making of management decisions [49]. Fluorescence measurements are also used to assess chlorophyll content. Fluorescence is related to photosynthetic activity; however, it is limited to an area of 100 cm$^2$ and optimized only for planophyll leaves [50]. There are a variety of sensors available for hyperspectral, multispectral, and fluorescence measurements, and examples of these instruments are given in Table 2.

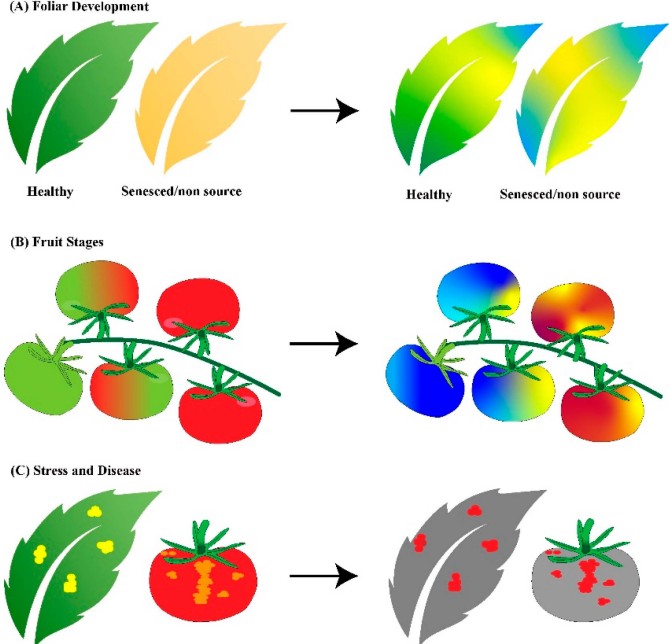

**Figure 3.** Demonstration of hyperspectral imagining applications. Hyperspectral imagining can be used to determine (**A**) foliar developmental stage, (**B**) fruit stage and ripening, and (**C**) stress and disease presence. The graphics on the left are an RGB color representation of what the human eye would see and the graphics on the right represent false color images after hyperspectral image processing.

Plant chlorophyll content is a measure of plant health; it also correlates with carotenoids, nitrogen, and maximum green fluorescence. One commonly used, well-tested technique for ascertaining plant chlorophyll content is the normalized difference vegetation index (NDVI). By using near infra-red (NIR), NDVI (NIR reflectance−red reflectance]/[NIR reflectance + red reflectance) can be calculated to estimate plant chlorophyll concentrations [23]. Scientists are developing other more efficient ways to measure plant chlorophyll content, using Google Glass. A leaf is put into a portable illuminating device, and two photographs are taken: one under white LEDs and the other under red LEDs. The two photos are sent to the server in under 10 s and analyzed for chlorophyll content. To date, they have successfully calibrated the equipment for 15 species. While these species were deciduous trees, this method could easily be applied to horticultural crops for the rapid assessment of plant health in the future [51]. Much of plant-health and fruit-quality monitoring is performed by using light sensors, and smartphone cameras are starting to become more useful in this realm. Smartphones have also been used to monitor plant stress, using the NIR spectrum, by evaluating NDVI. Smartphones have recently had their NIR-blocking filters removed, allowing for the sensing of NIR wavelengths by the CMOS sensor. Chung et al. (2018) used an NIR high-pass filter, which allowed for the sensing of wavelengths above 800 nm to collect NIR reflectance, and then the capture of red reflectance was performed without the filter [52].

### 5.4. Plant Disease

Plant diseases cause deleterious effects on the growth and development of crops, and such effects can reduce yields significantly and make the resulting agricultural products unfit for consumption. Globally, plant disease accounts for 10% of reduction in yield [53]. Currently, we lack understanding of many plant–pathogen systems and of the physiological mechanisms of disease symptoms in response to pathogen infection [53]. By using spectral images, which can measure light outside of the visible spectrum, we can quantify disease symptoms invisible to the human eye. Expanding the detection range may allow for earlier detection of diseases, allowing growers to take prompt action to mitigate the disease impact [6].

Plants emit a large array of volatile organic compounds (VOCs), which consist of various chemical classes, such as terpenes, fatty acid derivatives, alcohols, alkanes, alkenes, esters, isoprene, and acid. Plants emit these compounds from their flowers, fruits, leaves, and roots. Constitutive VOCs are those that are largely controlled by genetic and environmental conditions, and induced VOCs are those that are highly phenotypically plastic and affected by abiotic and biotic factors. Real-time VOC detection methods such as gas chromatography–mass spectrometry (GC–MS) and proton transfer reaction–mass spectrometry are starting to be used to investigate disease and pest presence on crops before visible indications are apparent [54]. Figure 4 demonstrates how, in the future, real-time measurement of VOC emissions from various plant organs could be performed by autonomous robots, while also executing crop harvests and maintenance tasks, such as pruning. These real-time measurements will help inform crop-management decisions that can be actioned before the abiotic or biotic stress is detrimental to crop health or negatively impacts yield. These methodologies can also be applied to fungal crop infections. Fungal infections impact crop production, and, for many infections, they are already widespread and difficult to treat by the time they are finally visible to the naked eye. By using continuous air-sampling techniques with optical sensors to detect the presence of spores, the detection of fungal infection can happen well before crop performance is impacted [45].

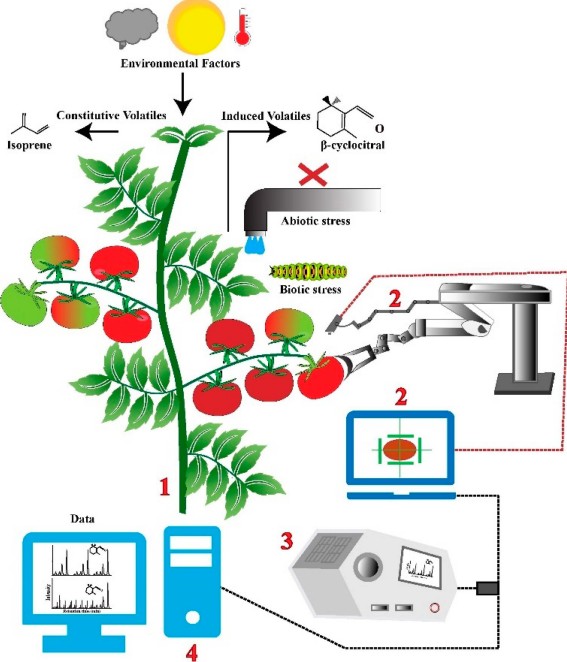

**Figure 4.** Graphic of future ability to understand abiotic and biotic stress by real-time measurement of VOC emissions through the use of autonomous robots and GC–MS measurement. A crop (1) can have VOC emissions monitored, while crop harvest and maintenance tasks are performed by an autonomous robot (2), with these VOCs measured with GC–MS (3) and further analyzed with the use of a computer (4) to understand VOC type and concentration. These measurements will inform crop-management actions to maintain a healthy crop with high yields.

*5.5. Breeding, New Varieties and Seeds*

Much of the advancement in plant phenotyping has been driven by breeding programs [24]. Research is now aimed toward producing varieties of plants through breeding that will be better adapted to low-input agriculture and resource-limited environments, with pest and disease resistance and drought tolerance. Seed selection is an important element within breeding programs, as seed germination rates and vigor are the two most important measurements for seed performance and, thus, plant performance. HTP seed phenotyping hardware and software technology are being developed that use machine-learning image-based technology to assess germination rates and vigor. This technology has been prototyped and tested on a number of crops and will be made commercially available in the near future [55].

Overall, by connecting a plant's genetic makeup, its genotype, to the internal and external characteristics expressed, its phenotype, plants can be selected for high yield and stress tolerance more rapidly, advancing breeding programs and maintaining quality fruit production over a crop cycle. Increasing breeding efficiency is hugely important for producing high-yielding and disease-tolerant varieties; however, phenotyping is important for monitoring in situ plant health and fruit quality in order to make optimal real-time adjustments [23]. Accurate phenotyping will help breeders select plants that will adapt to resource-limiting environments and low-input agricultural systems [56].

*5.6. Summary for Non-Destructive Plant Phenotyping*

Non-destructive phenotyping techniques are being developed, and some have been validated to assess crop growth, yield, fruit and leaf health status, and disease presence. While many of these techniques are under testing in scientific experiments, some of these techniques are still in the initial phases of development. Phenotyping imaging techniques have been validated for specific plant varieties; however, more data are needed to validate more varieties that can be phenotyped by using these technologies [23]. The data collected needs to be across growth stage, health status, and abiotic and biotic stresses so that specific image signatures can be defined for each plant variety. To validate such a range of plants will require huge amounts of data capture and analysis, which will require cross-disciplinary collaboration with artificial intelligence (AI) and machine learning (ML).

**6. Conclusions and Recommendations for Protected Cropping**

With the adoption of environmental control systems and rapid real-time phenotyping, a grower can maximize yields, improve aesthetic appeal, and reduce losses related to abiotic and biotic stresses. This transformation will likely increase the grower's competitive edge in emerging markets for customized, nutritious, and provenance-verified quality foods. To help advance the protected-cropping industry, it is recommended that growers of different scales implement the use of environmental sensors, climate-control mechanisms, and phenotyping techniques. These techniques will greatly assist in maximizing yield, reducing disease impacts, and hastening breeding programs to produce new varieties suited for future climates and environments. Fully integrating smart control of the growth facility will also increase flexibility of the grower to manage crops remotely.

Standardization and data management are key topics for the future of crop phenotyping. Big-data management and protocols will be necessary, as many of the phenotyping techniques explored herein require the collection of vast amounts of data, particularly imaging techniques, which require sophisticated postprocessing procedures that include self-learning algorithms [41]. Data will be generated that can be used to build libraries for ML [57]. With investment from experts in and developers of AI and ML, the postprocessing of large datasets can be achieved quickly, with inbuilt management-decision suggestions [58]. The development of the Internet of Things technology tailored to indoor cropping will greatly facilitate data transfer and analysis that can then be interpreted to make crop-management decisions [59].

**Author Contributions:** C.R.M. wrote the review, with inputs and revisions by D.T.T., Z.-H.C., O.G. and C.I.C. All authors have read and agreed to the published version of the manuscript.

**Funding:** The review was based on a report commissioned and funded by the Future Food Systems Cooperative Research Centre, which supports industry-led collaborations between industry, researchers, and the community. We also received financial support from Horticulture Innovation Australia projects (Grant Number VG16070 to DT, ZHC, OG, and CIC; Grant Number VG17003 to DT, ZHC; Grant Number LP18000 to ZHC) and CRC project P2-013 (DT, ZHC, OG, and CIC).

**Data Availability Statement:** Data for Figure 1 was reported from Graeme Smith's 2011 presentation: An Overview of the Australian Protected Cropping Industry. The presentation can be found here: https://www.graemesmithconsulting.com/images/documents/An%20Overview%20of%20 the%20Australian%20Protected%20Cropping%20Industry%20Compatibility%20Mode.pdf (accessed on 14 September 2022).

**Acknowledgments:** The authors highly appreciate Sidra Anwar for producing the graphics for Figures 3 and 4.

**Conflicts of Interest:** The authors declare no conflict of interest.

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
