# Peer review of "Precise Phenotyping for Improved Crop Quality and Management in Protected Cropping: A Review"

_2673-7655, doi:10.3390/crops2040024_

Round 1

Reviewer 1 Report

The manuscript is devoted to a discussion of the possibility of transferring the experience accumulated in the phenotyping of outdoor crops to protected conditions. Existing approaches that could be used for protecting cropping are discussed.

The manuscript presents the authors' reasoning and speculation as to what could be used. The authors provide no confirmation for their assumptions. Many of the approaches discussed are still in the research phase, and their practical applicability has not yet been proven.

Some of the approaches look controversial. For example, the authors' claim that EC can be used as an indirect assessment of nutrient content in soil (line 138 of the manuscript) is debatable. These possibilities are highly dependent on the properties of the particular soil and the type of nutrients.

Despite these shortcomings, it would be helpful to correct the text by removing the emphasis on Australia. The reasoning given is not only valid for this continent.

Author Response

We would like to thank reviewer 1 for their careful consideration while reading our article.  Below we address each point of Reviewer 1’s comments.

Reviewer 1:

Is the work a significant contribution to the field?  ***

Is the work well organized and comprehensively described?  ***  

Is the work scientifically sound and not misleading?  ***

Are there appropriate and adequate references to related and previous work?  ***       

Is the English used correct and readable?  *****

Comments and Suggestions for Authors

The manuscript is devoted to a discussion of the possibility of transferring the experience accumulated in the phenotyping of outdoor crops to protected conditions. Existing approaches that could be used for protecting cropping are discussed. The manuscript presents the authors' reasoning and speculation as to what could be used. The authors provide no confirmation for their assumptions. Many of the approaches discussed are still in the research phase, and their practical applicability has not yet been proven.

Although some of the techniques described in the review have not been completely vetted in protected cropping, they have been used in outdoor farming systems. We now clearly define whether these approaches have been used in protected cropping (PC) or whether they are still in the experimental stage for PC. This review provides scientists and growers some options to explore and perhaps trial in their facilities, with greater justification for these assumptions.

Some of the approaches look controversial. For example, the authors’ claim that EC can be used as an indirect assessment of nutrient content in soil (line 138 of the manuscript) is debatable. These possibilities are highly dependent on the properties of the particular soil and the type of nutrients.

We agree that EC in soils depends on soil properties. However, EC is ideal for hydroponic systems which were the focus of this statement. We have made that distinction clear in the review.

Despite these shortcomings, it would be helpful to correct the text by removing the emphasis on Australia. The reasoning given is not only valid for this continent.

We agree and have revised the manuscript to present a more global approach.

----------------------------------

We would like to thank Reviewer 2 for their careful consideration while reading our article.  Below we address each point of Reviewer 2’s comments.

Reviewer 2 Report

The review article “Precise phenotyping for improved crop quality and management in protected cropping: A Review” shows the current climate monitoring technology and the destructive and non-destructive phenotyping strategies developed for protected cropping to advance the Australian protected cropping sector. The paper has been studied and unfortunately found that ideas, write-up and the concept is presented in a poor way. This quality is very low from the major suggestions as there is much more to be performed than the normal improvements. Adding, nothing new has been presented into this article, the presented material is already available on different research domains in the shape of articles in of thousands. What is the innovation of this article? I could not find anything while studying it!

Moreover, what is Protected cropping (PC)? PC is not a recognized abbreviation, how authors can write a non-recognized abbreviation? Is this abbreviation approved from any respected government, agency, institute or country? In this case, artificial intelligence (AI) and machine learning (ML) are successfully recognized abbreviations, this is clear. But protected cropping (PC) not! Complete article is based on PC, not correct!

1. Abstract: Poor quality and quantity. Half of the abstract is based on the background till line 18, while the other half shows the general perspective of the industry. The industry is a news as it appeared “suddenly”: no background, no theme, nothing. The paper is about protected cropping, precise phenotyping and crop quality improvement, but I could not see anything relevant here.

2. Introduction: Not optimum: very low in quality and quantity, no idea is backed up, concept is not properly presented. Readers don’t understand the major topics, such as protected cropping, precise phenotyping and crop quality improvement after reading this section. Literature not enough.

3. Overview: The section seems like it will provide a comparison of the cropping systems in Australia with other countries of the world, but in reality, it’s not the case. Again, not background is given for these ideas in this section. Table 1: References? Literature analysis? Theme? Figure 1: Same?

4. Section 3: Phenotyping for advancement: how the authors have advanced or provided new ideas to advance the Phenotyping method for particular crops? I could study nothing innovative, creative and/or anything that could be at least interesting.

5. Section 4: Environmental monitoring: The parameters, such as Phenotypic plasticity, Root-zone temperature etc., Light quality and quantity, Temperature and relative, and Sensor technology, what is new here? Just browse these names in a simple search and you will find hundreds of articles with the same sentences. Nothing creative is provided here.

6. Table 2: References? Literature analysis? Theme?

7. Section: Its ok, there are some good things about this section. However, Figure 2 is a non-original figure with reference, but does not contribute anything new.

8. Figure 3 and Figure 4: Does not contribute anything new to the field! Idea is old, just a copy of the literature.

9. Summary: Does not make any sense! Summary of the paper or the section?

10. References: Poor: not enough. It’s a review article not a research paper, you must compare it with other articles.

Author Response

We would like to thank Reviewer 2 for their careful consideration while reading our article.  Below we address each point of Reviewer 2’s comments.

Reviewer 2:

Is the work a significant contribution to the field?  *

Is the work well organized and comprehensively described?  *

Is the work scientifically sound and not misleading?  *

Are there appropriate and adequate references to related and previous work?  *

Is the English used correct and readable?  ***

Comments and Suggestions for Authors

The review article “Precise phenotyping for improved crop quality and management in protected cropping: A Review” shows the current climate monitoring technology and the destructive and non-destructive phenotyping strategies developed for protected cropping to advance the Australian protected cropping sector. The paper has been studied and unfortunately found that ideas, write-up and the concept is presented in a poor way. This quality is very low from the major suggestions as there is much more to be performed than the normal improvements. Adding, nothing new has been presented into this article, the presented material is already available on different research domains in the shape of articles in of thousands. What is the innovation of this article? I could not find anything while studying it!

We appreciate the perspective of Reviewer 2.  The objective of this review was to assemble the known phenotyping techniques used in outdoor cropping and provide them to the protected cropping (PC) community. While sensor technology is abundant and evolving rapidly, it is commonly presented in piece-meal with single sensors or sensor systems described in detail. Our objective was to pull this information into a single document accessible to the PC community. Here, we discuss key areas identified by the industry, FAO and OECD to increase agricultural food security and improve crop quality by real-time, precise phenotype monitoring. 

As identified by Reviewer 1, we have attempted to summarize what is known in sensing and imaging of crops in the field and then identify technologies that can be applied in protected cropping. This is an emerging area of phenotyping and imaging. Therefore, we believe that our review presents a new angle and that is timely and important to the readers of Crops.

Moreover, what is Protected cropping (PC)? PC is not a recognized abbreviation, how authors can write a non-recognized abbreviation? Is this abbreviation approved from any respected government, agency, institute or country? In this case, artificial intelligence (AI) and machine learning (ML) are successfully recognized abbreviations, this is clear. But protected cropping (PC) not! Complete article is based on PC, not correct!

We understand that different terms are applied to crop production under cover in different countries. In Australia, protected cropping (PC) is the term used to describe growing plants under cover, in controlled or uncontrolled environmental conditions. The term was fully described in the manuscript, and it should be noted that the term is also recognized by the FAO. However, if the term is confusing, then we can spell out protected cropping in the article. Unfortunately, the reviewer did not provide a term or abbreviation that could be used to replace protected cropping. What term would you like us to use?

  1. Abstract: Poor quality and quantity. Half of the abstract is based on the background till line 18, while the other half shows the general perspective of the industry. The industry is a news as it appeared “suddenly”: no background, no theme, nothing. The paper is about protected cropping, precise phenotyping and crop quality improvement, but I could not see anything relevant here.

We have revised the Abstract to more fully describe protected cropping, precise phenotyping and crop quality improvement. The Abstract has been updated to better reflect the theme and organization of the manuscript.

  1. Introduction: Not optimum: very low in quality and quantity, no idea is backed up, concept is not properly presented. Readers don’t understand the major topics, such as protected cropping, precise phenotyping and crop quality improvement after reading this section. Literature not enough.

We have substantially added more information about our main topics of protected cropping, precise phenotyping, and crop quality to properly guide the reader to our main themes. We have also added more literature (+17 references) to more fully represent the breadth and depth of this field.

  1. Overview: The section seems like it will provide a comparison of the cropping systems in Australia with other countries of the world, but in reality, it’s not the case. Again, not background is given for these ideas in this section. Table 1: References? Literature analysis? Theme? Figure 1: Same?

We agree with the comment that the paper was too focused on Australia and led the reader to think that we would compare the Australian system with those in other parts of the world. We have broadened our paper to be inclusive of world cropping systems and provided sufficient background and literature to support this approach. We think that the themes are more clearly identified and discussed than in the original review.

  1. Section 3: Phenotyping for advancement: how the authors have advanced or provided new ideas to advance the Phenotyping method for particular crops? I could study nothing innovative, creative and/or anything that could be at least interesting.

This review focusses on providing an overview of available sensing and imaging techniques that are used in outdoor systems but might be modified or used in protected cropping. It should be noted that many of these technologies are still being developed.

  1. Section 4: Environmental monitoring: The parameters, such as Phenotypic plasticity, Root-zone temperature etc., Light quality and quantity, Temperature and relative, and Sensor technology, what is new here? Just browse these names in a simple search and you will find hundreds of articles with the same sentences. Nothing creative is provided here.

This review was intended to assemble current information on sensor technology and phenotyping into a single article so that growers using protected cropping systems could evaluate the potential for use in their system. It was not intended to provide in-depth evaluation or novel designs of any single sensor, which is more commonly presented in the “hundreds of articles” cited by the reviewer. Unfortunately, the reviewer was looking for an approach which was not the goal of this review paper, and hence the disappointment expressed by this reviewer.

  1. Table 2: References? Literature analysis? Theme?

We agree that the number of references was insufficient for this review paper. We have now added substantially more references that address the key themes of the paper.

  1. Section: Its ok, there are some good things about this section. However, Figure 2 is a non-original figure with reference, but does not contribute anything new.

Figure 2 has been placed here as an example of two different types of imaging:  Red Green Blue (RGB) and thermal imaging.  This review is to be a common place for both scientists and growers to find information about sensor technology and phenotyping techniques and we feel that this image provides a representative example of how RGB and thermal imaging can be used to understand plant health and water status.

  1. Figure 3 and Figure 4: Does not contribute anything new to the field! Idea is old, just a copy of the literature.

Figures 3 is used here as an example of ways in which hyperspectral imaging can be implored to identify various foliar development, fruit stage and stress or disease on crops.  Figure 4 is used here to show the reader a future vision of how advanced real-time phenotyping can be used to autonomously grow, maintain, and harvest produce. While the concepts are not new to science, our intention with both of these figures is to provide the reader with examples of how precise phenotyping can be used to improve crop quality and increase food security by reducing labor needs.

  1. Summary: Does not make any sense! Summary of the paper or the section?

We have clarified this section to reduce confusion.  There are two summary sections within the manuscript:  4.6 Summary and 5.6 Summary.  Each of these summaries refer to the section that preceded the summary.  We have changed the text to explicitly clarify this for each summary section.

  1. References: Poor: not enough. It’s a review article not a research paper, you must compare it with other articles.

We thank the reviewer for pointing this out.  We have added 17 references by adding more detailed information about the topic and expanding the subject to encompass the global industry. We have also more explicitly compared this review to other papers in this area.

Reviewer 3 Report

Dear Authors,

I really appreciated your article; both the objectives and the description of the different tools and techniques are clearly described and presented and I think that it could be very useful for the experts of this sector. I think there is only something to review regarding editing, images etc. please find below my suggestions.

·       In general, please check the spaces among words because sometimes they are missing, sometimes there is a double space. Please also check the lines within the text, because sometimes there are too many empty lines, as for example from line 410 to 418 and from line 428 to 473, and in other cases, as line 66-67 (table 1) perhaps a line of space should be given between the table and the text, and the same is for line 293, I would leave some space between the text and the figure.

·       line 202: smart phone should be smartphone.

·       Table 2: please give a definition of TDR and PAR, even at the end of the table.

·       Line 253: please define RGB.

·       Lines 358-359: terpenes are repeated twice.

·       line 419: but is it necessary to put this link that leads to table 3?

·       Table 3: the first image covers the text, perhaps it should be better adapted to the available space, and this applies to all images in the table. Furthermore, I think that the numbering of the photos should starts again from 1.

·       Conclusions: please evaluate if the subparagraph title can be deleted (just the title), in any case I think that AI and ML can be put in full here.

Author Response

We would like to thank Reviewer 3 for their careful consideration while reading our article.  Below we address each point of Reviewer 3’s comments.

Reviewer 3:

Is the work a significant contribution to the field?  ***** 

Is the work well organized and comprehensively described?  *****           

Is the work scientifically sound and not misleading?  *****         

Are there appropriate and adequate references to related and previous work?  ****     

Is the English used correct and readable?  ***** 

Comments and Suggestions for Authors

I really appreciated your article; both the objectives and the description of the different tools and techniques are clearly described and presented and I think that it could be very useful for the experts of this sector. I think there is only something to review regarding editing, images etc. please find below my suggestions.

In general, please check the spaces among words because sometimes they are missing, sometimes there is a double space. Please also check the lines within the text, because sometimes there are too many empty lines, as for example from line 410 to 418 and from line 428 to 473, and in other cases, as line 66-67 (table 1) perhaps a line of space should be given between the table and the text, and the same is for line 293, I would leave some space between the text and the figure.

  • line 202: smart phone should be smartphone.

Thank you for noticing this; it has been amended.

  • Table 2: please give a definition of TDR and PAR, even at the end of the table.

TDR has been replaced with moisture probe.  PAR was defined on line 195.

  • Line 253: please define RGB.

Thank you for catching this oversight.  RGB has been defined.

  • Lines 358-359: terpenes are repeated twice.

Thank you for catching the repetition – it has been deleted.

  • line 419: but is it necessary to put this link that leads to table 3?

We have changed the link that refers to Table 3.

  • Table 3: the first image covers the text, perhaps it should be better adapted to the available space, and this applies to all images in the table. Furthermore, I think that the numbering of the photos should starts again from 1.

We have made sure that the image is not covered by text.

  • Conclusions: please evaluate if the subparagraph title can be deleted (just the title), in any case I think that AI and ML can be put in full here.

The subparagraph has been deleted and AI and ML have been defined on line 495.

Round 2

Reviewer 2 Report

REV-1: The review article “Precise phenotyping for improved crop quality and management in protected cropping: A Review” is reviewed again. Unfortunately, it has not been improved as it was been asked previously. Changing words and grammar is not going to raise the bar: this quality is very low from the major suggestions as there is much more to be performed than the normal improvements. Nothing new is still added into this second version of the paper. Review is one of the most difficult paper to write, it could not be written in this way. Also, there is no take away message for the readers!!!

Abstract length is not optimum, very low quality, and quantity. What new things are added into the Introduction now?? I could not find anything new!!! Its more confused than the previous one. This paper does not provide/contribute anything new to the journal or science. There is no new thought in it. Authors need to understand that writing is not the only thing that makes a paper published , but a new and innovating concept that must be presented in an efficient way makes it more appealing and convincing to publish. I don’t think if a paper like this one which has super major improvements could be presented to the general readers in this very bad shape!!!

Thank you.